# A virus-encoded type I interferon decoy receptor enables evasion of host immunity through cell-surface binding

Bruno Hernáez[1], Juan Manuel Alonso-Lobo[1], Imma Montanuy[1], Cornelius Fischer[2,3], Sascha Sauer[2,3], Luis Sigal[4], Noemí Sevilla[5] & Antonio Alcamí[1]

Soluble cytokine decoy receptors are potent immune modulatory reagents with therapeutic applications. Some virus-encoded secreted cytokine receptors interact with glycosaminoglycans expressed at the cell surface, but the biological significance of this activity in vivo is poorly understood. Here, we show the type I interferon binding protein (IFNα/βBP) encoded by vaccinia and ectromelia viruses requires of this cell binding activity to confer full virulence to these viruses and to retain immunomodulatory activity. Expression of a variant form of the IFNα/βBP that inhibits IFN activity, but does not interact with cell surface glycosaminoglycans, results in highly attenuated viruses with a virulence similar to that of the IFNα/βBP deletion mutant viruses. Transcriptomics analysis and infection of IFN receptor-deficient mice confirmed that the control of IFN activity is the main function of the IFNα/βBP in vivo. We propose that retention of secreted cytokine receptors at the cell surface may largely enhance their immunomodulatory activity.

---

[1] Centro de Biología Molecular Severo Ochoa, Consejo Superior de Investigaciones Científicas (CSIC)-Universidad Autónoma de Madrid (UAM), 28049 Madrid, Spain. [2] Max Planck Institute for Molecular Genetics, 14195 Berlin, Germany. [3] Max Delbrück Center for Molecular Medicine (BIMSB/BIH), Robert-Rössle-Str. 10, 13092 Berlin, Germany. [4] Department of Microbiology and Immunology, Thomas Jefferson University, Philadelphia, PA 19107, USA. [5] Centro de Investigación en Sanidad Animal; Instituto Nacional de Investigación y Tecnología Agraria y Alimentaria, 28130 Valdeolmos, Madrid, Spain. Correspondence and requests for materials should be addressed to A.Aí. (email: aalcami@cbm.csic.es)

The control of primary virus infections in vertebrates often relies on an efficient type I interferon (IFN-I) response. IFN-I is a family of proinflammatory cytokines that are early induced during infection and mainly secreted from infected cells after recognition of viral products by pattern recognition receptors (PRRs). Among their pleiotropic effects, IFN-I (α and β) transmits its signals in healthy cells surrounding the site of infection through the type I IFN receptor (IFNAR) and the Jak/Stat signalling pathway to further initiate the transcription of several interferon stimulated genes (ISGs) that collaborate at diverse levels to establish an antiviral state and limit viral replication and spreading[1–4].

To overcome the IFN host response, most viruses have developed diverse strategies aiming to (i) minimise IFN induction, (ii) block IFN signalling and/or (iii) neutralise the antiviral activity of some ISGs[5–7]. Orthopoxviruses (OPVs) provide good examples of IFN evasion. While IFNs are required for protection against OPV infections[8–10], most members exhibit multiple, and often redundant, mechanisms to diminish the antiviral action of IFN (reviewed elsewhere[11–14]). Among these strategies, the secretion of soluble proteins that bind IFN-I with high affinities and prevent its interaction with IFNAR provides an efficient and straightforward way to counteract this response[15–17]. Interestingly, although these viral IFN-I binding proteins (IFNα/βBPs), that serve as decoy receptors, do not share sequence similarity with the cellular IFNAR, they are highly conserved among prominent members of this virus genus, such as Variola virus (VARV), the aetiologic agent of smallpox and one of the most aggressive pathogens faced by humankind, monkeypox virus (MPXV) which also results pathogenic for humans, some strains of Vaccinia virus (VACV), the smallpox vaccine, Ectromelia virus (ECTV), the causing agent of mousepox, or cowpox virus (CPXV)[15–18].

The well-characterised secreted IFNα/βBP from VACV, named B18, lacks a transmembrane domain but is found at the cell surface. This protein was identified as the soluble early antigen from VACV, a protein detected in supernatants and the surface of infected cells[19]. The IFNα/βBP binds to the surface of uninfected cells in the surrounding tissue, thus preventing the IFN-mediated induction of an antiviral state before cells become infected[17,20]. This ability was confirmed for the VARV and MPXV B18 orthologues[16]. Using a site-directed mutagenesis approach with VACV B18 and its VARV and MPXV orthologues, we demonstrated that binding to the cell surface is mediated by glycosaminoglycans (GAGs) and occurs through conserved clusters of basic residues located at the amino terminus of these proteins[21]. The interaction of these IFNα/βBPs with GAGs at the cell surface does not interfere with their ability to bind IFN-I, since mutant proteins that failed to attach to the cell surface retained the ability to bind and block IFN-I with high affinity[21]. In the case of ECTV, a basic cluster at the amino terminus of its IFNα/βBP is also present and, although binding to GAGs has not yet been addressed, this IFNα/βBP has been recently detected on the cell surface in the spleen and liver of infected mice[21,22].

Poxvirus IFNα/βBPs are considered essential for virus virulence, and its key contribution to poxvirus pathogenesis was demonstrated in two different mouse models by using viruses lacking IFNα/βBP expression. Deletion of the B18R gene from VACV caused a 100-fold attenuation in intranasally infected Balb/c mice[15]. Moreover, a stronger attenuation (>10^7-fold) was obtained for ECTV, since inactivation of the IFNα/βBP gene resulted in an avirulent variant. Importantly, in contrast to the full lethality of the wild type (WT) ECTV, all the animals survived after footpad inoculation with the IFNα/βBP deletion mutant[23]. However, the biological relevance of the cell surface binding properties of the poxvirus IFNα/βBP has never been determined in the infected animal host.

ECTV, as occurred with VARV and humans, has a narrow host-range and infection of mice represents a model of coevolution of a virus with its natural host, and it is considered of special interest to study viral immune evasion mechanisms[24,25]. ECTV inoculation of susceptible mouse strains, such as Balb/c, results in systemic lethal mousepox, which shares with human smallpox multiple properties: signs of disease, high mortality rates and the ability of VACV to protect from infection[25–29]. These common features support the value of mousepox as a model for human smallpox.

ECTV usually enters its host through abrasions in the skin and after initial replication it migrates to the draining lymph node. Via efferent lymphatic vessels and blood, the virus reaches major target organs (spleen and liver) where it massively replicates. If the animal survives, a second hematogenous spread from these sites to the skin produces rash and pocks characteristic of poxviral diseases, such as smallpox and monkeypox.

In the present study we have addressed the aforementioned question regarding the specific contribution of the cell surface binding ability of the IFNα/βBP to poxvirus pathogenesis. Using both VACV and mousepox infection models, we demonstrate in vivo that the interaction with GAGs is essential for the IFNα/βBPs to effectively block the IFN-I antiviral effects during poxvirus infections.

## Results

**Cell surface binding of the ECTV IFNα/βBP.** We have demonstrated that B18, the IFNα/βBP from VACV, binds GAGs at cell surface, and that three basic regions (A, B and C) in the N-terminus of this viral protein are required for cell interaction, but not for binding and blocking IFN-I (Fig. 1a and ref. [21]). Based on previous data and the conservation of basic motifs at the amino terminal region of the ECTV IFNα/βBP[21,22], we hypothesised that E194, the B18 orthologue from ECTV Naval strain (named E166 in the Moscow strain), behaves like B18. To test this, we expressed C-terminal V5-6xHis tagged versions of E194 and E194^GAGmut in a baculovirus system. E194^GAGmut aims to mimic the previously generated B18^GAGmut, originally named IM15 (Fig. 1a and ref. [21]), so that seven basic residues (R or K) from region A, which is located at the Ig1 domain of E194, were replaced by alanine (A) (Fig. 1a).

After protein purification, we incubated equal amounts of E194 and E194^GAGmut with Chinese hamster ovary K1 (CHO-K1) and HeLa cells, and binding was determined by immunofluorescence and flow cytometry using anti-V5 antibodies. As negative control we used an unrelated ECTV protein, a V5-6xHis tagged version of the poxvirus encoded semaphorin (SEMA). Examination after incubation with E194 revealed a membrane pattern in both cell lines, which was not observed with E194^GAGmut or SEMA (Fig. 1b). Flow cytometry analysis in CHO-K1 cells showed a similar histogram for E194 and B18, confirming that the changes introduced in E194 to create E194^GAGmut completely abolished its ability to bind to the cell surface (Fig. 1c). We conclude that E194 attaches to the cell surface, that region A is involved in GAG binding and that GAGs presumably mediate such interaction.

We next evaluated whether these changes affected the E194 capacity to bind IFN-I and block its antiviral effects. We first determined by surface plasmon resonance (SPR) in a BIAcore biosensor the kinetic binding parameters of E194 to mouse IFN-α and compared it to those of E194^GAGmut. In both cases, the recombinant WT protein and its mutant version, the calculated $K_D$ was in the nanomolar range indicating a high affinity for IFN-I. These results are consistent with those previously determined for the interaction of VACV B18 and its B18^GAGmut derivate with human IFN-I[15,21] (Fig. 2b). Additionally, we determined that E194 efficiently binds human IFNα and β but with reduced affinity in the case of mouse IFNβ (Supplementary Figure 1), as described for B18[15] and coincident with previous data using supernatants from ECTV infected cells[18].

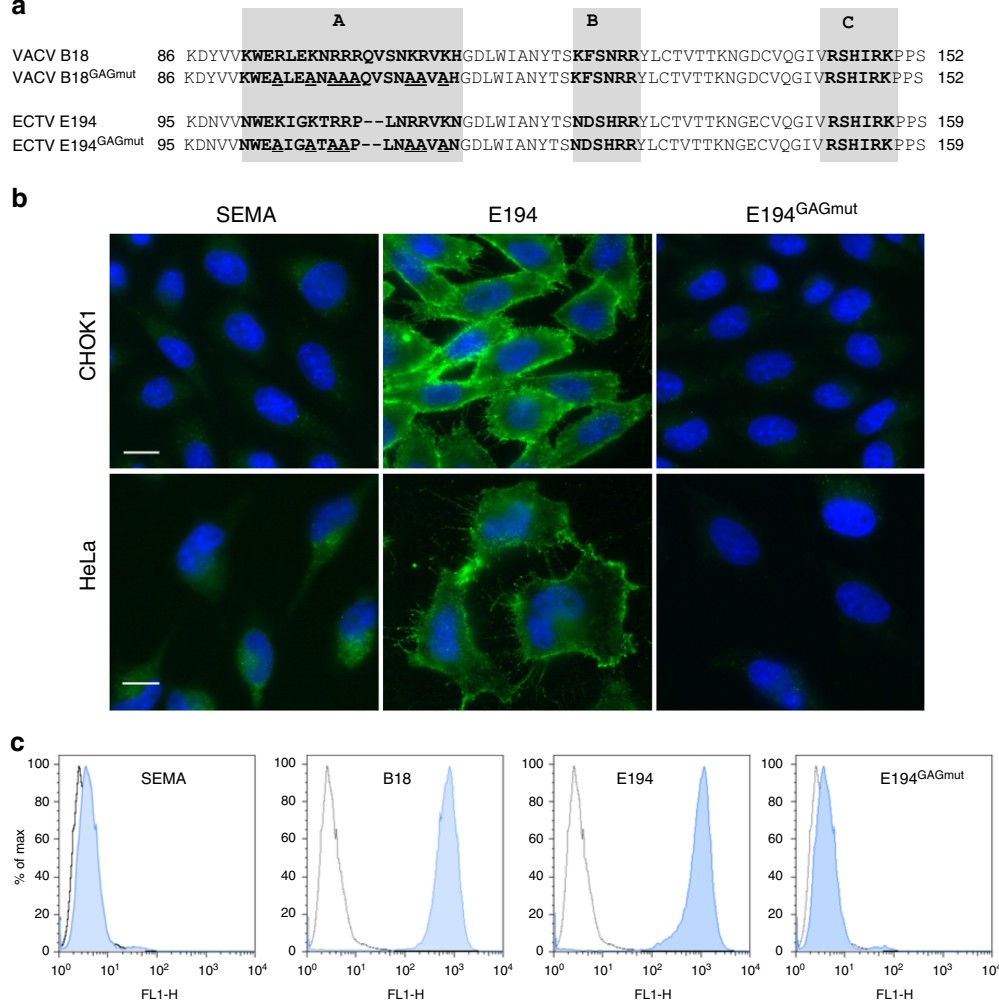

**Fig. 1** Cell surface interaction of the IFNα/βBP from ECTV. **a** Partial amino acid sequence of the IFNα/βBP from VACV (B18) and ECTV (E194). Shaded boxes indicate the putative GAG binding regions A, B and C identified in the amino terminal sequence of the proteins. Amino acid substitutions of basic residues performed to generate their corresponding mutant proteins B18GAGmut and E194GAGmut are underlined. **b** The indicated cells were incubated at 4 °C for 30 min with V5-tagged recombinant proteins E194 and E194GAGmut, and then extensively washed. Proteins were detected by immunofluorescence with an anti-V5 monoclonal antibody and cell nuclei were stained with DAPI. The unrelated viral protein SEMA was used as negative control. **c** CHOK1 cells were incubated with recombinant tagged proteins on ice and extensively washed, and the binding to cell surface was assessed by flow cytometry using an anti-V5 antibody. VACV B18 protein was incorporated as positive control. Shaded histograms correspond to the indicated protein while non-shaded correspond to untreated cells. In every case, experiments were performed in triplicate and a representative histogram is shown

We then tested the ability of E194 to block the IFN-induced antiviral cell response. As expected, IFN treatment protected cells from vesicular stomatitis virus (VSV) infection (Fig. 2b, left panel). In contrast, preincubation of IFN with increasing amounts of E194, similar to B18, diminished the protective effect of IFN and cells died after VSV challenge in a dose-dependent manner (Fig. 2b, left panel). This capacity was not affected by the introduced changes abolishing its cell surface binding ability, since E194GAGmut also blocked the IFN-mediated cell protection (Fig. 2b, left panel). On the other hand, using a modified version of this assay, in which cells were first incubated with recombinant viral proteins and extensively washed prior the IFN addition, we found that E194GAGmut was no longer able to counteract the IFN protective effect against VSV infection (Fig. 2b, right panel). WT B18 and E194 proteins were retained at the cell surface, whereas the E194GAGmut protein was removed after extensive washing of the monolayer and no longer inhibited the IFN effects. In summary, we conclude that the GAG-binding properties described for the VACV IFNα/βBP[21] are extended to the ECTV IFNα/βBP.

**Viruses expressing an IFNα/βBP unable to interact with GAGs.** IFNα/βBP has been already shown to be relevant for VACV and ECTV virulence in mouse models of infection[15,23]. Aiming to discriminate the specific relevance of the interaction of the IFNα/βBP with GAGs at cell surface during poxvirus infections we generated mutant VACV and ECTV expressing the GAG-binding deficient versions of the IFNα/βBP. The VACV IFNα/βBP deletion mutant (VACVΔB18), referred here as VACVΔIFNα/βBP, was previously described[15] and we constructed a new ECTV IFNα/βBP (E194) deletion mutant in the Naval strain, referred to here as ECTVΔIFNα/βBP, since previous work has been done with ECTV Moscow strain[23]. We introduced in the IFNα/βBP gene locus of the deletion mutant viruses the corresponding sequences encoding variant IFNα/βBPs lacking the ability to bind to the cell surface, generating the recombinant viruses VACVIFNα/βBPGAGmut and ECTV IFNα/βBPGAGmut (Supplementary Figure 2). The genome sequence of the recombinant viruses constructed was fully sequenced to confirm the correct deletion or insertion of genes and the absence of additional mutations.

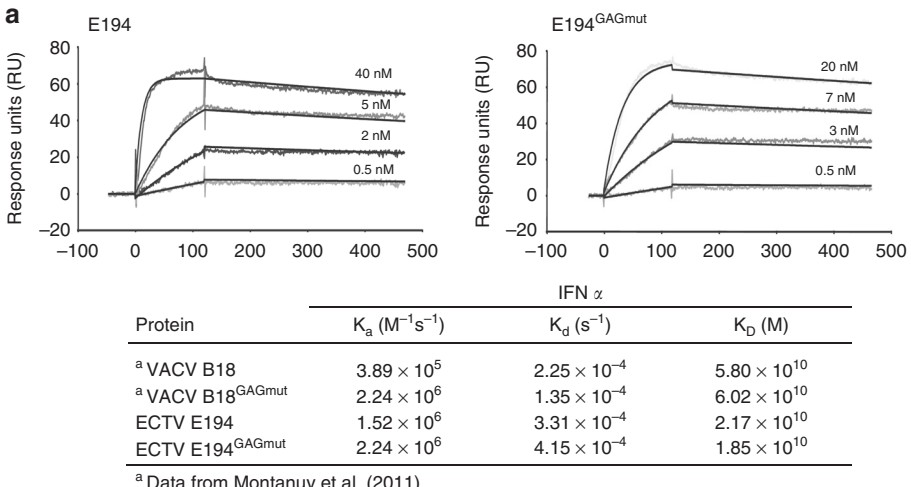

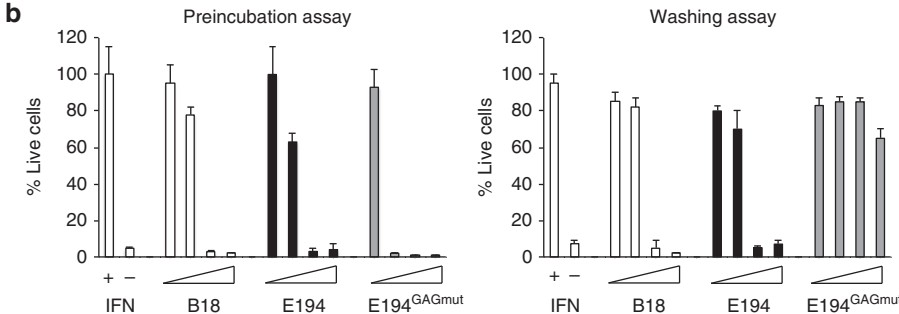

**Fig. 2** IFN-I binding properties of the variant E194 protein unable to interact with GAGs. **a** SPR sensorgrams and fitting obtained for the determination of the kinetic constants of the interaction of E194 and E194^GAGmut recombinant proteins with mouse IFNαA. E194 or E194^GAGmut were immobilised in a SPR Biacore SA sensor chip and binding and dissociation of several concentrations of mouse IFNαA at 30 µl/min were recorded and adjusted to a 1:1 Langmuir fitting (solid lines). The nanomolar concentration corresponding to each sensorgram is indicated. Kinetic parameters and calculated affinity constants of E194 and E194^GAGmut are shown in the inset and compared to those previously described for the interaction of human IFNα-2b with VACV B18 and B18^GAGmut (originally named IM15). **b** To determine the IFN-I blocking activities, in the preincubation assay (left), increasing amounts of recombinant proteins were incubated with 50 U of mouse IFNαA and the mixture added to L929 cells. After 16 h incubation, cells were infected with VSV and cell viability measured at 72 hpi. In the washing assay (right), cells were incubated with recombinant proteins, extensively washed, and then incubated with 50 U of mouse IFNαA as described for the previous assay. In both assays, untreated (IFN -) or IFN-I treated (IFN +) cells, and also the addition of B18 recombinant protein were incorporated as controls. Data are means showing standard deviations of two independent experiments performed in triplicate

To assess whether VACV and ECTV replication in cell culture was affected by the genomic changes, BSC-1 cells were infected with the recombinant viruses at low m.o.i. (0.01 pfu/cell), and virus yields determined by plaque assay at diverse times postinfection. No significant differences were found in the replication of the mutant viruses when compared to WT viruses (Fig. 3a, left panels). Similar results were obtained after infection of cells at high m.o.i. (5 pfu/cell), since no differences were found in the production of cell-associated virus and extracellular virus at 24 hpi (Fig. 3a, right panels). These results showed that expression of the variant form of IFNα/βBP does not affect virus replication in cell culture.

We analysed the supernatants from cells infected with these mutant viruses to test their capacity to block the IFN-induced protective effects against VSV (Fig. 3b). As expected, preincubation of IFN-I with supernatants from cells infected with VACVΔIFNα/βBP and ECTVΔIFNα/βBP did not prevent the IFN-mediated protection. On the contrary, supernatants from infections with viruses expressing the IFNα/βBP^GAGmut, VACVIFNα/βBP^GAGmut and ECTVIFNα/βBP^GAGmut, acted like those from their corresponding WT viruses and were able to prevent the protective action of IFN-I against VSV (Fig. 3b, left panels). However, in the washing assay we found that supernatants from cells infected with VACVIFNα/βBP^GAGmut and ECTVIFNα/βBP^GAGmut were unable

to block the protective effect of IFN-I against VSV, while those from WT viruses still retained the IFN-blocking capacity (Fig. 3b, right panels). Together these results indicated that (i) all viruses replicate similarly in cell culture, (ii) the WT and variant forms of the IFNα/βBPs are not required for infection in cell culture, and (iii) the IFNα/βBP^GAGmut versions are correctly synthesised by the engineered viruses during infection.

**IFNα/βBP binding to cell surface contributes to virulence**. To evaluate the contribution of the interaction of the IFNα/βBPs with cell surface GAGs to poxvirus virulence we used two models of infection. In the case of VACV, we used the intranasal (i.n.) infection model, where mice were infected with $10^5$ pfu of the WT and mutant viruses (Fig. 4). Coincident with previous results[15], all animals infected with WT VACV succumbed to infection, whereas infection with VACVΔIFNα/βBP did not cause any mortality and mice showed no weight loss and limited signs of illness (Fig. 4a). Interestingly, although animals inoculated with VACVIFNα/βBP^GAGmut showed clear signs of infection and weight loss, only one of these animals died (Fig. 4a) indicating an attenuated phenotype of VACVIFNα/βBP^GAGmut similar to that observed with the IFNα/βBP deletion mutant.

In the case of ECTV, we used the classical mousepox model of subcutaneous (s.c.) inoculation in the footpad, with viral

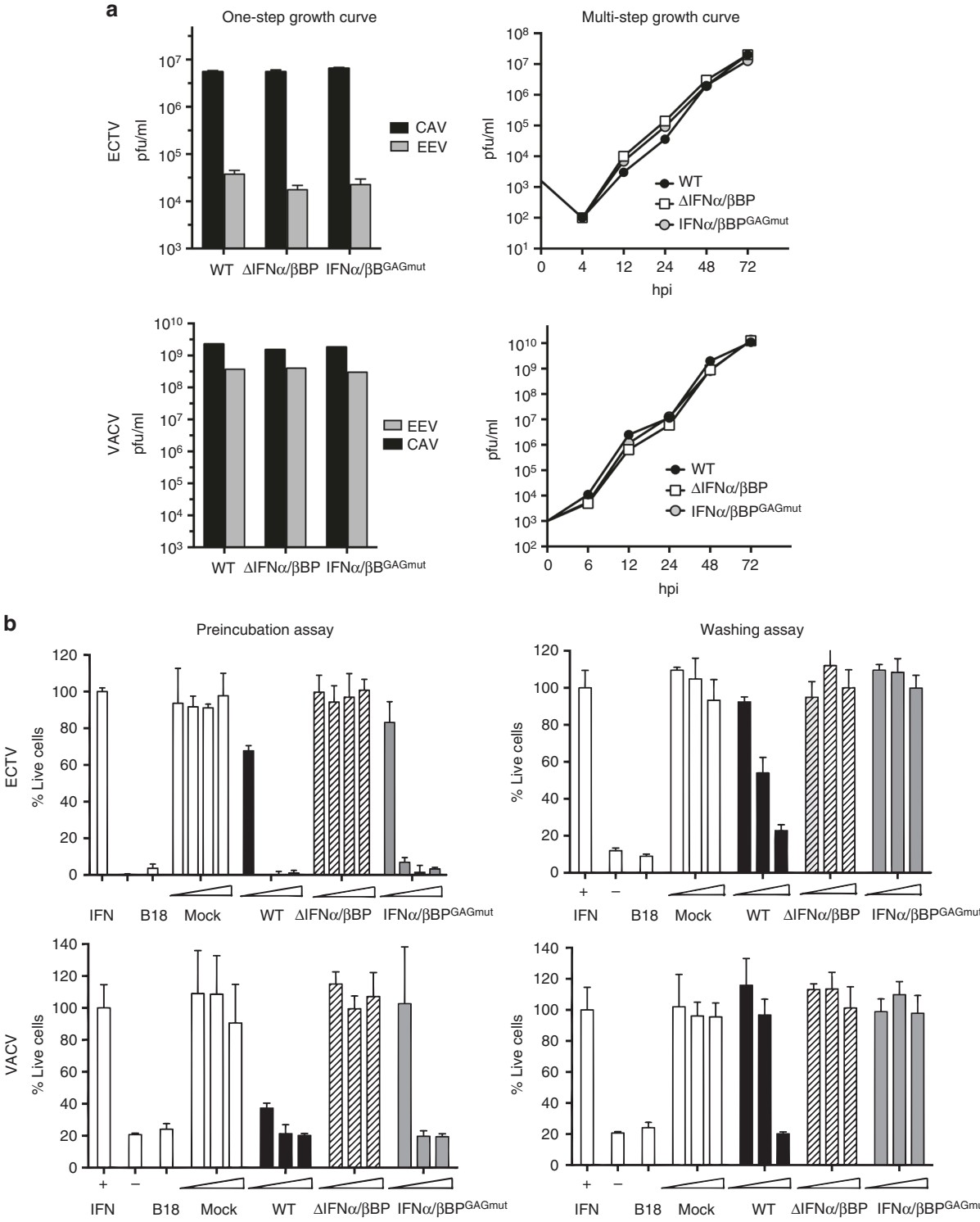

**Fig. 3** Characterisation of mutant viruses in cell culture. **a** BSC-1 cells were infected with the indicated viruses at 0.01 pfu/cell for multi-step growth curves or 5 pfu/cell for one-step growth curves. Total virus production was determined by plaque assay at the indicated times in the multi-step growth curve. Virus titres at 24 hpi from fractions containing cell associated viruses (CAV) or extracellular viruses (EV) where determined in the one step growth curves. **b** Detection of IFN-I blocking activity in supernatants from recombinant ECTV (upper graphs) or VACV (lower graphs) infected cells was performed using the previously described preincubation (left) or washing (right) assays. In both assays, controls of untreated (IFN -) or IFN-I treated cells (IFN +) and also B18 recombinant protein were incorporated as positive control. Data are means showing standard deviations of two independent experiments performed in triplicate

doses ranging from 10 to $10^6$ pfu. The inoculation of 10 pfu of WT ECTV resulted in 100% lethality by 10 dpi, whereas all the animals infected with ECTVΔIFNα/βBP survived the infection even with $10^6$ pfu (Fig. 4b). As observed with VACV, ECTVIFNα/βBP$^{GAGmut}$ was highly attenuated, even at the highest dose of $10^6$ pfu where only one animal died (Fig. 4b). These results showed that both IFNα/βBP$^{GAGmut}$-expressing mutant VACV and ECTV were highly attenuated to levels similar to those found with the corresponding IFNα/βBP deletion mutants.

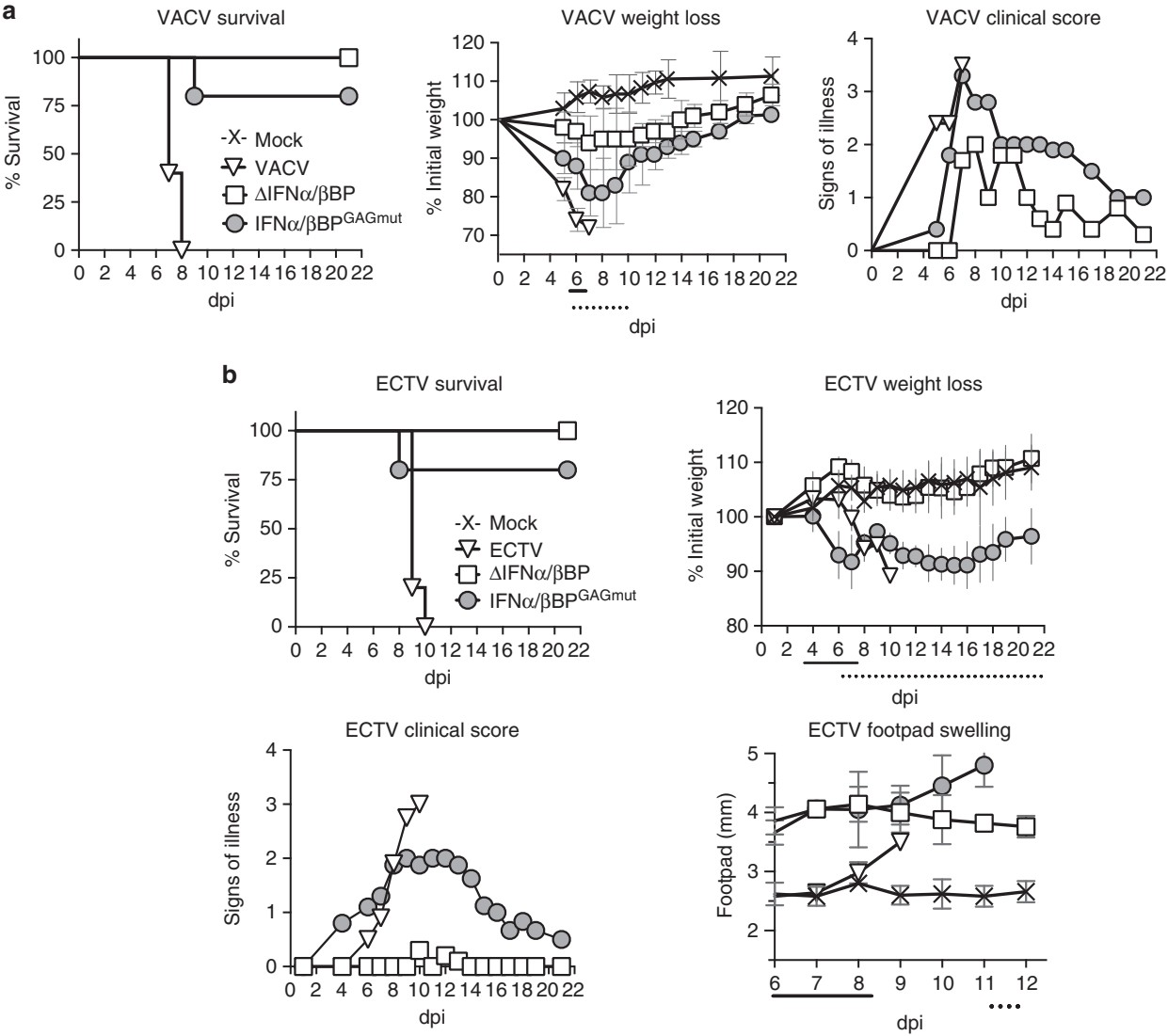

**Fig. 4** The IFNα/βBP cell surface binding activity contributes to poxvirus pathogenesis. Balb/c mice ($n = 5$) were infected i.n. or s.c. in the footpad with the indicated VACV ($10^5$ pfu) (**a**) or ECTV (10 pfu of WT and $10^6$ pfu of the IFNα/βBP mutants) (**b**), respectively. Mice were monitored daily for survival, bodyweight and signs of illness, and footpad swelling in the case of ECTV. Weight data are expressed as the mean ±SD of the five animals weights compared to their original weight at the day of inoculation. Signs of illness, as a score ranging from 1 to 4, and the size of the footpad (mm) where inoculation was performed are also expressed as the mean of the five animals. Statistical analyses performed were multiple $t$-tests with FDR $Q = 1\%$ and days at which significant differences ($p < 0.05$) were found between IFNα/βBP$^{GAGmut}$ and WT or ΔIFNα/βBP infections are indicated with solid or dotted lines, respectively. One representative experiment of two is shown

In the case of ECTV infections, an important difference regarding illness severity was observed when comparing the two modified viruses. Mice infected with the highest dose of ECTVΔIFNα/βBP did not exhibit weight loss below 95% of the original values or other signs of illness characteristics of mousepox, with the exception of a strong footpad swelling not observed in the WT virus infection. Interestingly, this increase in footpad swelling appeared along with a pronounced weight loss and severe signs of illness in mice infected with ECTVIFNα/βBP$^{GAGmut}$ (Fig. 4b, c). The magnitude of the disease produced by this mutant virus was dependent on the dose inoculated. While infection with low viral doses (10 or $10^2$ pfu) did not result in weight loss or significant signs of illness, mice infected with doses higher than $10^4$ pfu exhibited weight loss from 4–5 dpi to later drop to nearly 80% of their initial value, and illness scores peaked at 10–14 dpi at around 2.5 (scale: 0 for healthy animals and 4 for severely sick animals). Despite this severity of infection, all the

ECTVIFNα/βBP$^{GAGmut}$ infected mice started to recover from disease and increased bodyweight from 16 dpi (Fig. 4 and Supplementary Figure 3).

Collectively, these results evidenced that the ability of the IFNα/βBP to interact with the cell surface strongly contributes to poxvirus disease. To discriminate whether this attenuation of poxviruses expressing the IFNα/βBP$^{GAGmut}$ is due to IFN-I modulation or to an unknown function associated to the cell surface attachment of the IFNα/βBP, we evaluated the virulence of ECTV expressing the IFNα/βBP$^{GAGmut}$ in a host lacking IFN-I signalling. To this effect, mice deficient in subunit 1 of IFNAR (IFNAR1) were infected s.c. in the footpad with $10^2$ pfu. We found that viruses expressing either no IFNα/βBP or IFNα/βBP$^{GAGmut}$ recovered its virulent phenotype and killed all mice between 7 and 9 dpi, as observed with WT ECTV (Fig. 5). These results support the idea that the ECTVIFNα/βBP$^{GAGmut}$ attenuation in WT susceptible mice is due to a reduced IFN-I

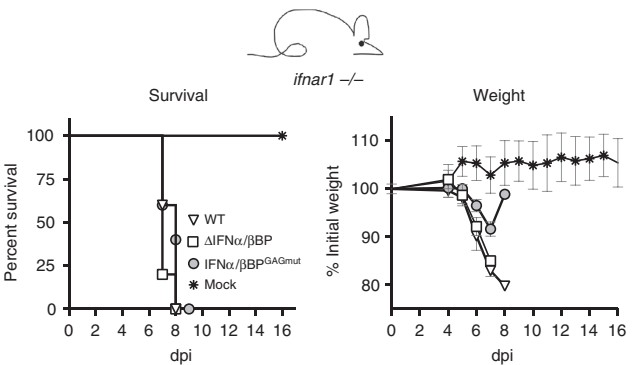

**Fig. 5** Infection of mice lacking IFNAR1. Mice deficient in IFN-I cellular receptor (*ifnar1* −/−) were s.c. inoculated in the footpad with 100 pfu of WT or the indicated mutant ECTV, and monitored daily for survival and weight loss. Mean ±SD data corresponding to one experiment with five mice per group is shown

blocking ability when the IFNα/βBP is unable to attach to the cell surface.

**IFNα/βBP cell surface attachment affects virus dissemination.** ECTV footpad inoculation mimics natural infections and is an excellent model to examine whether the IFNα/βBP interaction with the cell surface is required for in vivo virus spread within the infected animal. Following footpad inoculation of ECTV in susceptible mice, the virus replicates at the site of infection before reaching the popliteal lymph node. From this point, the virus disseminates via the efferent lymphatics and bloodstream to invade the spleen and liver, where massive replication occurs[30–32]. We infected Balb/c mice in the footpad with $10^3$ pfu of ECTV, ECTVΔIFNα/βBP or ECTVIFNα/βBP$^{GAGmut}$, and virus yields from footpad, spleen and liver were titrated at 5 and 7 dpi. Coincident with previous data[23], no differences among the three viruses were observed at both times postinfection in footpads, indicating that the IFNα/βBP function is not required for the initial virus replication (Fig. 6, left panels). However, important differences were found in major target organs. In the absence of the IFNα/βBP, viral titres at 5 dpi were 4 and 3 log lower in spleen and liver, respectively, compared to WT virus infection and titres in both organs were under the detection limit of the assay at 7 dpi (Fig. 6, central and right panels). In the case of ECTVIFNα/βBP$^{GAGmut}$, we observed a discrete reduction in viral titres at 5 dpi compared to WT ECTV, which was much more evident (3 log reduction) in both organs at 7 dpi, suggesting a role for IFN-I in virus clearance in these major target organs between 5 and 7 dpi (Fig. 6, central and right panels). The spleen and liver from ECTVΔIFNα/βBP infected mice exhibited a healthy appearance, similar to that observed in mock-infected mice, and completely different to the reduced size and necrotic aspect characteristic of WT ECTV infections. However, the spleen and liver from the ECTV-IFNα/βBP$^{GAGmut}$-infected animals showed an unhealthy intermediate appearance between ECTVΔIFNα/βBP- and ECTV-infected animals, with limited liver damage and enlarged spleens (Supplementary Figure 4).

The above data indicated a restriction in virus dissemination from the initial site of replication to major target organs after infection with ECTVIFNα/βBP$^{GAGmut}$. Thus, to ascertain whether an efficient blocking of IFN-I facilitates poxvirus replication in these major target organs, we circumvented the natural route of ECTV infection by intravenous (i.v.) inoculation of mice. Intravenous inoculation of $5 \times 10^3$ pfu of ECTV resulted in the early appearance of signs of illness (3 dpi) and weight loss, leading to 100% mortality with a mean time of death of 8 dpi (Fig. 7). As found in the footpad model of infection, ECTVΔIFNα/βBP was

highly attenuated in the i.v. model and mortality was importantly diminished (30%) after inoculation with ECTVIFNα/βBP$^{GAGmut}$, showing an extended mean time of death of 10 dpi. Again, similar differences to those described above in terms of disease severity were appreciated here, since the signs of disease resulted more evident in the ECTVIFNα/βBP$^{GAGmut}$ -infected than in the ECTVΔIFNα/βBP-infected animals (Fig. 7). Additionally, after i.v. inoculation with lower doses ($5 \times 10^2$ pfu) 37% of the animals survived to WT infection, whereas no deaths were observed with both ECTVΔIFNα/βBP and ECTV-IFNα/βBP$^{GAGmut}$. These data indicated that an efficient IFN-I blocking mediated by IFNα/βBP and its cell attachment ability is critical for poxvirus replication in the major target organs.

**Expression of a cell-surface mutant IFNα/βBP increases systemic anti-IFN activity.** Binding to GAGs might allow retention of the poxviral IFNα/βBP to locally increase its IFN blocking function at sites of virus replication. Thus, abrogation of the cell surface attachment properties of the IFNα/βBP may result in increased systemic dispersion of the anti-IFN activity in the bloodstream during infection. To evaluate this hypothesis we tested the IFN protection activity in plasma from infected animals at 5 dpi. The levels of protection against IFN determined in the case of VACV expressing the IFNα/βBP$^{GAGmut}$ were significantly increased compared to those found in the WT infection (Fig. 8a), indicating a higher prevalence of the IFNα/βBP in serum after abrogation of its GAG binding ability. We found more variability in ECTV-infected animals, although the data again suggested a higher prevalence in plasma from mice infected with a virus expressing the mutant version of the IFNα/βBP compared to infection with a virus expressing the WT protein (Fig. 8b). In this case, four out of five ECTVIFNα/βBP$^{GAGmut}$ -infected animals analysed exhibited higher protection levels against IFN than the mean value from WT ECTV infections.

**Lack of IFNα/βBP cell binding elicits an IFN host response.** To determine whether the lack of cell surface attachment properties in the IFNα/βBP directly impacts on the IFN-induced host response, differential expression analyses using RNA-seq were undertaken on cervical lymph nodes, spleen and lung tissues from mice mock-infected or intranasally infected with VACV or the IFNα/βBP mutant viruses. Comparison of host gene transcription in mock-infected mice with that from mice infected with VACV, VACVΔIFNα/βBP or VACVIFNα/βBP$^{GAGmut}$ identified significant differentially expressed genes (SDEGs). Pathway enrichment analysis performed with those SDEGs retrieved a number of pathways related to the biological function of the IFN in those infected animals, including *I*FN signalling, Role of pathogen recognition receptors (PRRs) or Activation of IRF by cytosolic PRRs, among others (Fig. 9a).

Similar *z*-score values for these pathways were determined for VACVIFNα/βBP$^{GAGmut}$ and VACVΔIFNα/βBP infections in the lymph nodes, spleen and lung samples, indicating a clear and similar activation of these IFN related pathways. On the contrary, no activation (lymph node and spleen) or lower *z*-scores (lung) were determined for the WT infection (Fig. 9b). Importantly, upstream regulator analysis with splenic samples showed increased *z*-scores activation values for relevant transcription factors of the IFN response (IRF1, IRF3, IRF5, and IRF7), and also central genes in IFN signalling (IFN-I, IFNAR, and STAT1). However, the activation values for TRIM24, a negative regulator of the IFN/STAT pathway, were found diminished after infection with both IFNα/βBP mutant viruses compared to WT virus (Fig. 9c). Accordingly, 39.8% of the upregulated SDEGs (177/444) in VACVIFNα/βBP$^{GAGmut}$-infected spleens compared with WT infections were determined as ISGs regulated by IFN-I using Interferome 2.0[33]. Similar results were found in the other tissues

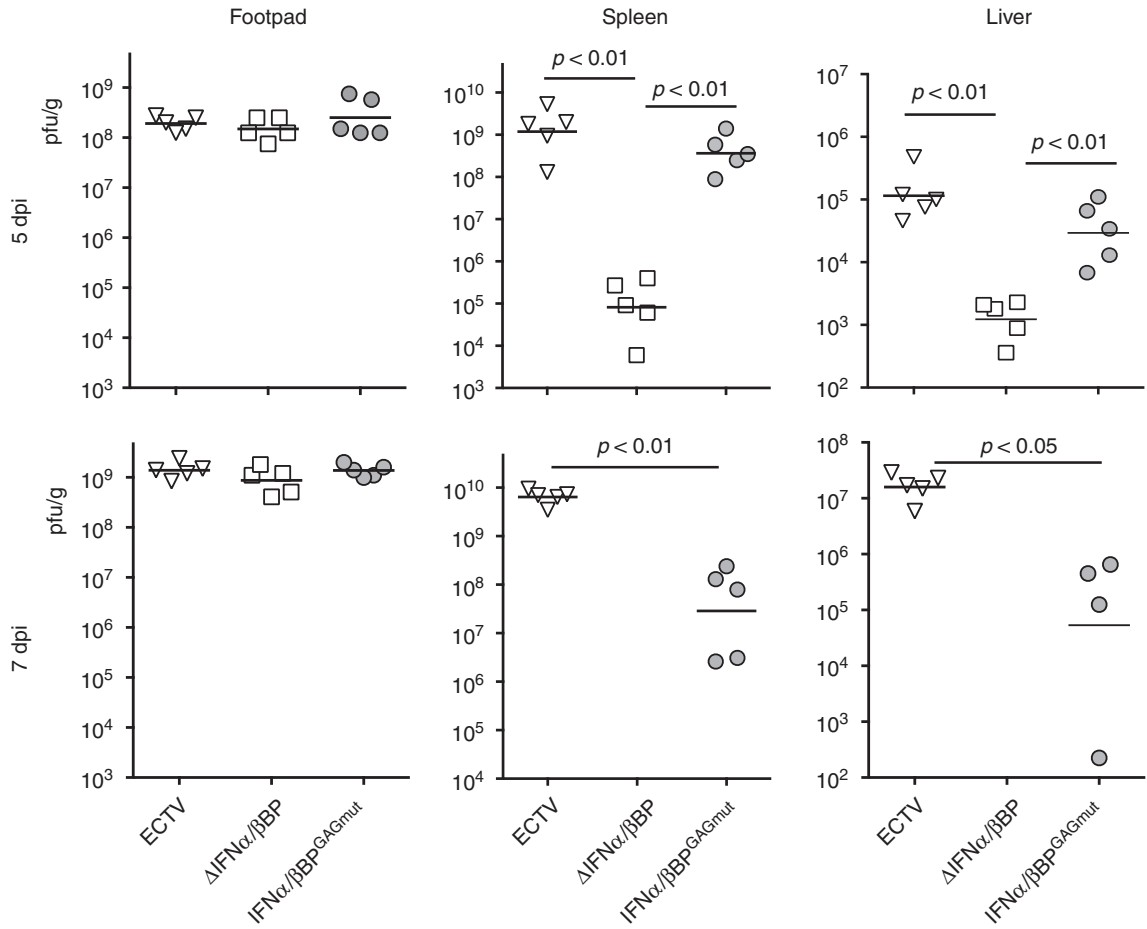

**Fig. 6** Virus dissemination in mice infected with ECTV mutants. Five Balb/c mice per condition were s.c. inoculated in the footpad with $10^3$ pfu of the indicated ECTV recombinants and virus titres in footpad, spleen and liver were determined by plaque assay at 5 and 7 dpi. Independent virus titres from each infected animal and the mean value are shown. Data below the detection limit of the assay ($10^2$ pfu/mg) are not depicted. Statistical significance indicated after Mann–Whitney test, assuming 0 value for undetectable viral loads

analysed, infected lymph nodes and lungs (Supplementary Table 1). Finally, expression levels of some selected ISGs were found increased after infection with VACVIFNα/βBP$^{GAGmut}$ and VACVΔIFNα/βBP, indicating a stronger IFN-induced response compared to WT VACV infection (Fig. 9d). These results indicated that the cell attachment properties in the IFNα/βBP are crucial to efficiently prevent the IFN induced host response during poxvirus infections.

## Discussion

The activity of the viral IFNα/βBP represents a potent and crucial immunomodulatory mechanism to evade the host innate immune IFN response during infection, and one of the most important virulence factors described to date for poxvirus infections of mice, since its absence results in up to $10^6$ fold attenuation[15,23]. Moreover, antibodies against this protein block its biological activity and efficiently protect mice from mousepox, confirming its contribution to virulence[22]. A similar degree of attenuation of ECTV has been recently reported after deletion of the gene encoding a viral homologue of the cellular tumour necrosis factor receptor, named cytokine response modifier D (CrmD), that inhibits TNF but also a set of chemokines through a unique chemokine binding domain, named smallpox virus-encoded chemokine binding (SECRET) domain[34]. These reports highlight the relevance of IFN-I, TNF and chemokines working together in the protective immune response against poxvirus

infections, similar to the link described for IFN-I and nuclear factor kappa B pathways[35].

The ability of the viral IFNα/βBP to attach to surfaces of non-infected and infected cells, where additionally exerts its inhibitory function, has long been known, however the specific contribution of this feature of the viral IFNα/βBP to pathogenesis remained undetermined[16,17,20]. We have addressed this question using engineered VACV and ECTV expressing mutant versions of the IFNα/βBP that no longer interact with the cell surface, and we demonstrate that this property of the viral IFNα/βBP is essential to prevent the IFN-I host response in both mouse models of infection. The attenuation phenotypes obtained for these mutant viruses were comparable to those of their correspondent IFNα/βBP deletion mutant viruses, revealing that binding IFN-I with high affinity is crucial, but not sufficient, to efficiently evade the IFN-I antiviral effects in vivo. This surprising result demonstrates that the localisation of the viral IFNα/βBP at the cell surface is required to efficiently fulfil its biological function in vivo.

As described for other GAG binding proteins, such as growth factors or chemokines[36,37], the binding to the cell surface may potentially confer diverse advantages to the viral IFNα/βBP during infection, functioning as a retention and clustering mechanism to act mainly within infected areas, for protection against circulating plasma proteases or to allow interactions with other molecules, in addition to IFN-I. In the case of the viral IFNα/βBP, our results using IFNAR deficient mice indicate that the main function of this protein at the cell surface is the

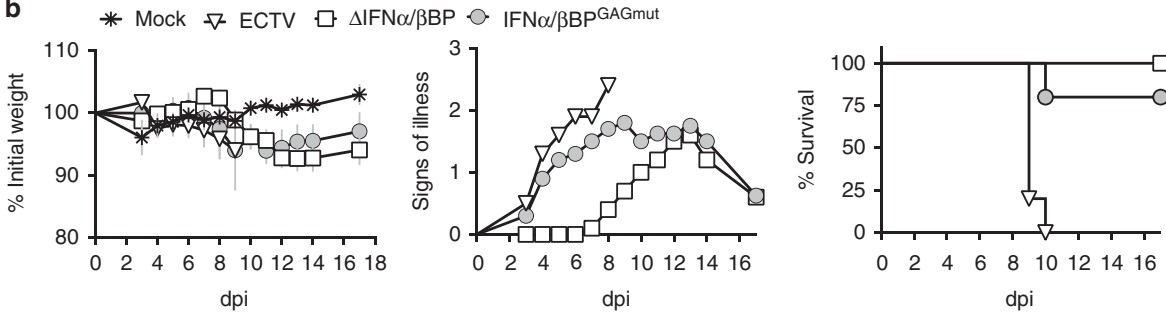

**Fig. 7** Infection by i.v. inoculation of ECTV IFNα/βBP mutants. Balb/c mice were i.v. infected with $5 \times 10^2$ or $5 \times 10^3$ pfu of the indicated ECTV mutants. **a** Survival rates and mean time to death (MTD)±SD from three independent experiments are shown. **b** A representative experiment of two showing weight loss, clinical signs and survival from animals infected with $5 \times 10^3$ pfu of the ECTV mutants

prevention of IFN-I signalling. In agreement with this, previous transcriptomics (RNA-seq) studies showed the absence of host gene expression changes after binding of the viral IFNα/βBP to a mouse cell line[38]. Thus, the benefits obtained after attachment to the cell surface would be restricted to the local enhancement of the inhibitory function of the viral IFNα/βBP and the abrogation of its GAG binding properties leads to a protein which is no longer able to control the IFN-I response during infection. This decrease in IFN-I inhibitory function in cell culture would explain that infection of mice with both the deletion mutants and the IFNα/βBP[GAGmut] expressing viruses resulted in a dramatic attenuation phenotype compared to WT virus infections. The IFN-I activation in mice infected with the IFNα/βBP[GAGmut] expressing viruses is expected to enhance T (CD8[+], CD4[+]) and NK cell responses, as shown in mice infected with an ECTV IFNα/βBP mutant[23]. The interaction of the IFNα/βBP with GAGs may interfere with attachment of chemokines to the cell surface, as proposed for some viral CKBPs[39]. Although the IFNα/βBP will mask a reduced number of GAG-binding sites at the cell surface, we cannot rule out the possibility that the binding of IFNα/βBP to GAGs may have some indirect effect on chemokine-mediated leucocyte migration to sites of infection.

Virus dissemination is significantly impaired after infection in the absence of a fully functional viral IFNα/βBP. Viral loads in major target organs from animals infected with the IFNα/βBP deletion mutant virus dropped to undetectable levels between 5 and 7 dpi, suggesting a role for the IFN system in virus clearance that would support the lack of evident mousepox disease. In the case of the sole abrogation of the viral IFNα/βBP cell attachment properties, the viral loads in spleen and liver, although clearly reduced compared to WT infection, still remained detectable at 7 dpi indicating a loss of viral IFNα/βBP effectiveness to prevent the IFN effects. This level of viral replication, coincident with limited liver damage and enlarged spleens, may account for the signs of illness observed in mice infected with the IFNα/βBP[GAGmut] expressing viruses. In addition to this reduction of virus titres in major target organs, we detected by RNA-seq the activation of an IFN based response in spleen, comparable to that elicited by the infection in the absence of the viral IFNα/βBP, supporting the idea that after

abrogation of its cell attachment properties the IFNα/βBP is not able to control the IFN host response and the subsequent virus clearance from major target organs.

Attachment to cell surfaces through GAG interactions is not a unique feature of the viral IFNα/βBP, and the conclusions from the present study could be applied to other poxvirus secreted immunomodulatory proteins that also exhibit this ability[39]. The IL-18 binding protein from Molluscum contagiousm virus and VARV neutralises IL-18 induced-effects, either as soluble or membrane anchored form, acting as a decoy receptor as shown for the viral IFNα/βBP[40,41]. Other examples include E163 from ECTV and M-T1 from MYXV, chemokine-binding proteins able to bind simultaneously chemokines and GAGs at the endothelium cell surface[42–44], and the poxviral complement inhibitor from VACV and VARV, exerting its function at the cell surface after GAG binding[45,46]. In each case, the phenotype attributed to the virus deletion mutants in these immunomodulators resulted attenuated to different degrees, revealing their contribution to poxvirus pathogenesis[47–52], however there is no data concerning the specific contribution of their GAG-binding properties. In the case of VCP, the attachment to cell surface may also occur in a GAG-independent fashion, through disulphide bond interaction with the VACV transmembrane protein A56[53]. A VACV mutant expressing a variant of VCP lacking cell surface localisation resulted in an attenuated phenotype similar to that of the VCP deletion mutant after i.n. inoculation of mice, and produced intermediate sized lesions between the deletion mutant and the WT viruses in a intradermal model of infection[51]. But this study was not conclusive because the VCP mutagenesis performed affected not only the binding to the cell surface but also its dimerisation ability, which is known to potentiate the VCP action and is mediated by the same amino terminal-free cysteine. In this case, it seems difficult to discern the specific contribution of cell surface attachment to pathogenesis. Instead, the viral IFNα/βBP has been proved to be a suitable immunomodulatory protein to test the contribution of its cell attachment properties to poxvirus virulence.

IFN-I signalling has emerged as an important and central pathogenic factor in the course of a variety of diseases, especially

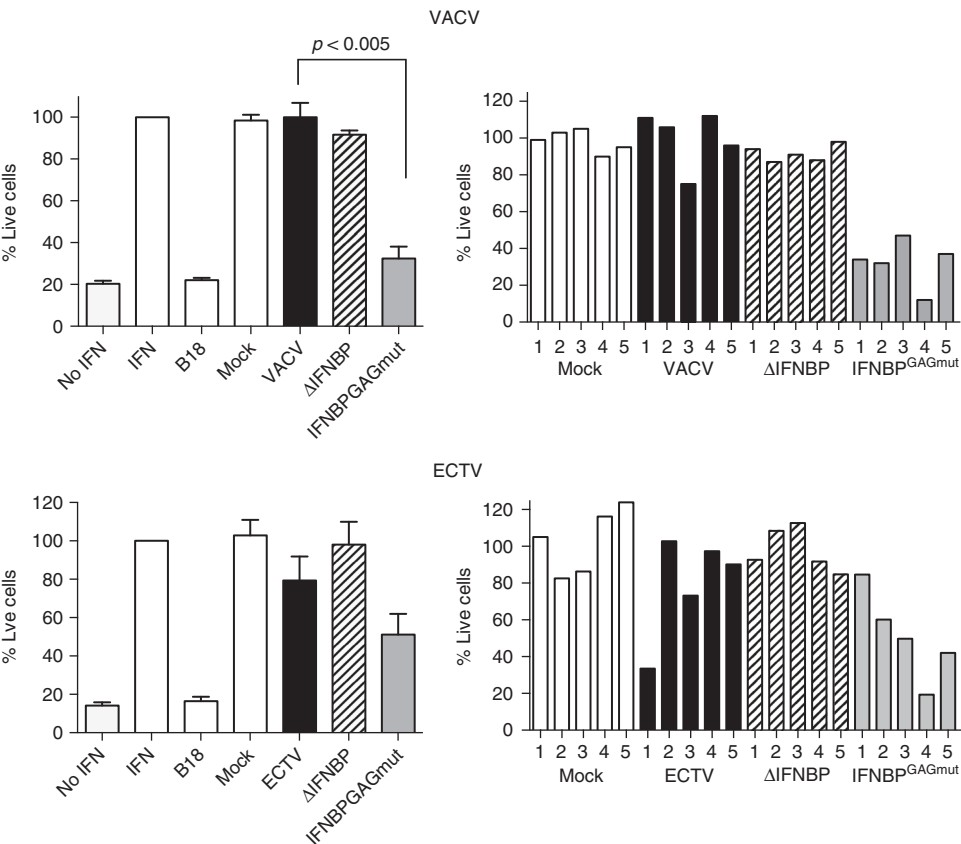

**Fig. 8** Detection and biological action of the IFNα/βBP in plasma from infected animals. Dilutions containing 20% of plasma from mice infected with the indicated viruses at 5 dpi were independently incubated with 50 U of human IFNαA for VACV samples or 50 U of mouse IFNαA for ECTV samples, and then added to HeLa or L929 cells, respectively. After 16 h cells were infected with VSV and cell viability measured at 72 hpi to determine the IFN blocking activity from plasmas. Mean±SEM from a representative experiment in triplicate from two with 5 mice/group is shown in the left panels, while the corresponding individual values obtained for each mouse are shown in the right panels

autoimmune and autoinflammatory disorders, such as systemic lupus erythematosus (SLE), type 1 diabetes or Sjogren's syndrome among others[54,55]. The current therapeutic approaches, some of them already in phase III clinical trials, for the treatment of these interferonopathies involve the targeting of IFN-I using either monoclonal antibodies against IFN-α or its receptor IFNAR, and even the induction of polyclonal anti-IFN-I neutralising antibodies after administration of inactivated cytokine derivatives or kinoids. However, in the case of SLE, these therapies based on monoclonal antibodies failed to completely subvert the IFN-I gene signature, which is a hallmark in SLE patients[56]. A possible explanation is that IFN-β and other IFN-α isoforms still remained active after treatment, and in this sense poxvirus IFNα/βBPs exhibit an additional advantage over monoclonal antibodies since they bind multiple forms of IFN-I[15,50]. In addition, the VACV IFNα/βBP has been recently shown to alleviate neurotoxicity and histopathological complications originated by the IFN-I upregulation in an HIV encephalitis mouse model and the viral soluble decoy receptor has been proposed as a viable therapeutic alternative to monoclonal antibodies against IFN-I[57,58]. We have demonstrated here that an engineered viral IFNα/βBP, lacking cell attachment properties while maintaining intact its IFN-I binding ability, is much less efficient than the unmodified version to prevent the host IFN-induced antiviral response during poxvirus infections. However, the GAG binding mutant may still represent a more potent inhibitor in anti-IFN therapeutics for these disorders, since it might potentially exert its inhibitory function at more distant tissues than the unmodified viral IFNα/βBP version. Of broader interest, the identification of GAG-binding activities in human secreted cytokine decoy receptors

or the addition of a GAG-binding domain deserves to be investigated in order to address whether the cell-binding activity may improve the therapeutic value of cytokine decoy receptors.

In summary, we show the critical role that the GAG-binding activity of a viral secreted IFN-I decoy receptor has on its ability to control the IFN response in vivo, illustrating the relevance of the cell surface binding activity. Our findings raise the possibility that the GAG-binding properties, either natural or engineered, of human soluble cytokine receptors may also be relevant to achieve their full immunomodulatory activity.

## Methods

**Cells and viruses**. BSC-1 (ATCC CCL-26), HeLa (ATCC CCL-2), Vero (ATCC CCL-81) and mouse L929 cells (ATCC CCL-1) were maintained in 10% foetal bovine serum (FBS) containing DMEM. CHO-K1 cells were kindly provided by Dr. Arenzana-Seisdedos (Institute Pasteur, Paris, France) and were grown in 10% FBS containing DMEM-Ham′s F12 (1:1) medium. Hi5 insect cells were cultured in TC-100 medium supplemented with 10% FBS The sources of virulent VACV Western Reserve strain and the corresponding VACV mutant lacking IFNα/βBP expression (VACVΔIFNα/βBP) were previously described[15]. The source of ECTV strain Naval was described[59]. VACV and ECTV were grown in BSC-1 cells, and viral stocks were partially purified by centrifugation through a 36% sucrose cushion and stored in Tris-HCl pH 9.0 at −80 °C. Viral stocks were titrated twice by plaque assay in monolayers of BSC-1 cells prior to animal infections. To this effect, serial dilutions of the virus stock were plated on duplicate BSC-1 cell monolayers in semi-solid carboxy-methyl cellulose media with 2.5% FCS. Cells were fixed in 10 % formaldehyde at 6 dpi and plaques were stained with 0.1% (w/v) crystal violet. VSV strain Cocal was obtained from Dr. William James (Sir William Dunn School of Pathology, Oxford University, Oxford, UK). Both VSV growth and titration by plaque assay were carried out in Vero cells.

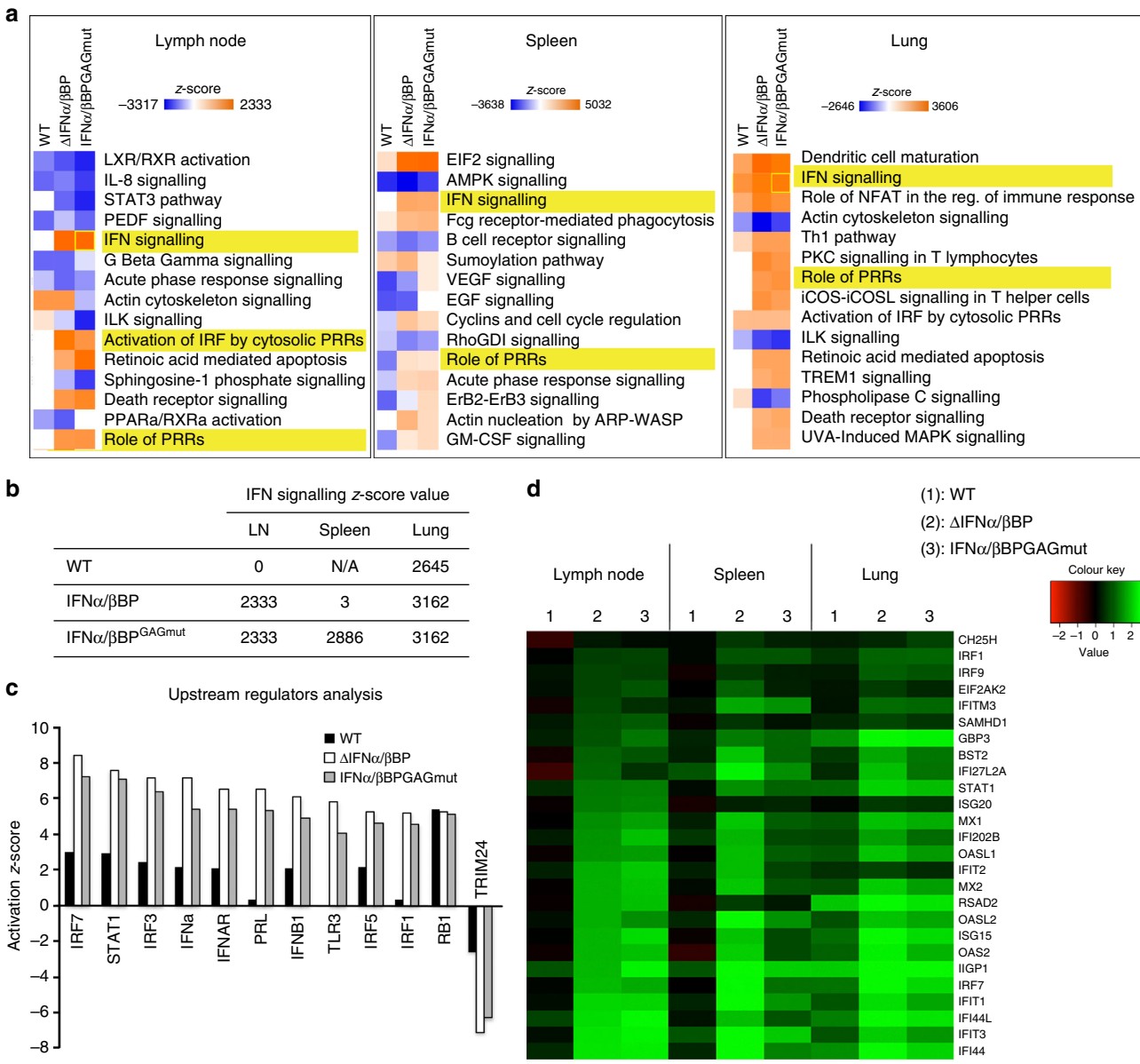

**Fig. 9** RNA-seq analysis of the IFN response after infection with VACV IFNα/βBP mutants. Two groups of five Balb/c mice per condition were i.n. inoculated with 10³ pfu of the indicated VACV. Differential expression analyses against mock-infected samples were performed using biological duplicates. **a** Heatmap showing the activation z-scores for 15 significantly enriched pathways using the SDEGs retrieved after comparison of each VACV infected sample with the mock-infected samples. **b** Activation z-scores determined for the Interferon Signalling pathway and corresponding to each VACV infected sample after comparison to mock infection. **c** SDEGs were analysed by the IPA upstream regulator function (direct only) and the activation z-score values for the selected upstream regulators determined for VACV-WT (black bars), VACVΔIFNα/βBP (white bars) and VACVIFNα/βBP^GAGmut (grey bars) infections are shown. **d** Heatmap showing the expression values (log2FC) of the selected ISGs in the indicated VACV infected tissues

**Construction of recombinant poxviruses.** Recombinant ECTVs were generated using ECTV-Naval as parental virus and a transient dominant selection procedure based on puromycin resistance to avoid the presence of foreign DNA sequences in the genome of the final recombinant ECTV[60]. Briefly, to generate ECTVΔIFNα/βBP, BSC-1 cells were infected with ECTV-Naval (0.01 pfu/cell) and at 1 hpi transfected with pBH13 using Fugene HD (Invitrogen), following the manufacturer indications. The plasmid pBH13 is based on pMS30 and contains a 927 nt fragment and a 883 nt fragment corresponding to 5′ and 3′ flanking regions of EVN194 gene (184,716 to 185,635 bp and 186,557 to 187,449 bp positions in the ECTV Naval genome)[59], together with a downstream cassette including the enhanced green fluorescence protein (EGFP) marker under the control of an OPV early/late synthetic promoter and the puromycin resistance gene. When cytopathic effect was complete, cells were harvested and used as inoculum in five consecutive rounds of infection in the presence of 10 μg/ml puromycin monitoring EGFP expression. ECTVΔIFNα/βBP was finally isolated by three successive plaque purification steps of white plaques in the absence of puromycin.

To generate ECTVIFNα/βBP^GAGmut a similar procedure was used, but this time with pBH17 as the vector and ECTVΔIFNα/βBP as parental virus. The plasmid pBH17 carries the sequence encoding E194^GAGmut, which had been first amplified by PCR from vector pIM26 (see below) and inserted between the previously indicated flanking regions in pBH13.

In the case of VACVIFNα/βBP^GAGmut, BSC-1 cells were infected with VACVΔIFNα/βBP[15] and transfected with pIM33. The plasmid pIM33 contains the nucleotide substitutions in *B18R* generated with the QuickChange II site-directed mutagenesis kit (Stratagene), using the oligonucleotides 5′-GGTTAAATGGGAAGCGCTAGAAGCAAATGCAGCGGCACAGGTTTCTAATGCAGCTGTTGCACATGTGATTTATGG -3′ and 5′-CCATAAATCACCATGTGCAACAGCTGCATTAGAAACCTGTGCCGCTGCATTTGCTTCTAGCGCTTCCCATTTAACC -3′ and the plasmid pAA21 as template[15]. The recombinant virus was isolated by the transient dominant selection procedure based on mycophenolic acid resistance mediated by the *E. coli gpt* gene as described[61]. A schematic genomic representation of the poxviruses used is shown in Supplementary Figure 2.

The genome from the recombinant VACV and ECTV generated was fully sequenced with the Illumina sequencing technology to confirm the genetic structure of the viruses and the absence of inadvertent mutations that may have been introduced during the generation of recombinant viruses.

**Virus growth curves.** BSC-1 cells were infected for 1 h at 37 °C at high m.o.i. (5 pfu/cell) or low m.o.i. (0.01 pfu/cell) in the one-step or multi-step growth curves, respectively. Cells were then washed and fresh medium was added. In the one-step growth curve, at indicated times post-infection, the medium was harvested and centrifuged at $1800 \times g$ for 5 min to pellet detached cells. These cells were combined with infected cells that had been scraped from the plate into 0.5 ml of fresh medium. In the multi-step growth curve, cells and media were together harvested at indicated times. In all cases, samples were frozen, thawed three times and titrated on BSC-1 cells in duplicates.

**Mice.** Balb/c female mice were purchased from Charles River. The IFNAR deficient mice (*ifnar1* −/−) on a C57BL/6 genetic background[9] were bred and maintained in the animal facilities of Instituto Nacional de Investigación y Tecnología Agraria y Alimentaria (INIA). All mice used in experiments were 5–8 weeks old.

**Ethical statement.** All animal experiments were performed with special efforts to minimise animal suffering, in compliance with national and international regulations and were approved by the Ethical Review Board of Consejo Superior de Investigaciones Científicas and Comunidad de Madrid under reference PROEX 025/16.

**Infection of mice and tissue processing.** The infections of groups of five animals per condition were performed with sucrose-purified virus diluted in PBS containing 0.1% bovine serum albumin. Mice were anaesthetised with isofluorane and then infected by s.c. inoculation into the footpad in the case of ECTV or i.n. infected in the case of VACV, with 10 μl of virus inoculum containing the indicated infectious doses. When indicated, i.v. infection of Balb/c mice with ECTV was performed by injection of 50 μl of virus inoculum containing $5 \times 10^3$ pfu into the tail vein. Mice were housed in ventilated racks under biological safety level 3 containment facilities and monitored daily for survival, weight and signs of disease.

For determination of virus titres in organs, infected mice were euthanized at 5 and 7 dpi by $CO_2$ asphyxiation, and a sample of the indicated organs from each animal were aseptically removed. Samples were weighed, homogenised in 1 ml PBS, and freeze/thawed 3 times. Then, the amount of infectious virus was determined by plaque assay as described above. For determination of anti-type I IFN activity in infected mice, plasma was prepared from nearly 1 ml of blood at 5 dpi by centrifugation at $14,000 \times g$ for 10 min at 4 °C.

**Expression and purification of recombinant proteins.** The recombinant baculoviruses expressing the VACV IFNα/βBP (B18) have been previously described[16,21]. The EVN194 coding sequence lacking its predicted signal peptide was PCR-amplified from the ECTV genome and cloned into pAL7, a modified pFastBac vector bearing the honeybee mellitin signal peptide at the 5′ region and a C-terminal V5-6xHis tag, to generate the plasmid pIM32. The plasmid pIM26 encoding E194$^{GAGmut}$ was engineered using pIM32 as template and introducing mutations in the E194 gene with the QuickChange II site-directed mutagenesis kit (Stratagene) and oligonucleotides 5′-GGTTAACTGGGAGGCGATAGGAGCG ACTGCGGCCCCTCTTAATGCAGCTGTTGCAAACGGTGACTTATGG -3′ and 5′-CCATAAGTCACCGTTTGCAACAGCTGCATTAAGAGGGGCCGCAG TCGCTCCTATCGCCTCCCAGTTAACC -3′. Plasmids pIM32 and pIM26 were sequenced to confirm the presence of the desired mutations and the absence of inadvertent mutations and then used to obtain the recombinant baculoviruses by the Bac-to-Bac system (Invitrogen). As control, a recombinant baculovirus expressing the ECTV SEMA protein C-terminally tagged to V5-6xHis was used. Supernatants from Hi5 cells infected with the recombinant baculoviruses were first concentrated by ultrafiltration and then buffer exchanged against 50 mM phosphate, 300 mM NaCl and 10 mM imidazole, pH 7.4, as described[21], prior to affinity-purification of recombinant proteins with Ni-NTA agarose (Qiagen). Protein purity and quantity were analysed on Coomassie blue-stained SDS-PAGE and quantified by gel densitometry.

**Protein binding to cell surface by immunofluorescence.** HeLa or CHO-K1 cells seeded onto glass coverslips in 24-well plates were incubated with 250 nM of purified recombinant viral proteins for 30 min at 4 °C. Cells were then extensively washed with ice-cold PBS and fixed with 4% paraformaldehyde in PBS for 12 min at room temperature. The membrane permeabilization step was omitted, thus after aldehyde quenching with 50 mM $NH_4Cl$ for 5 min, cells were incubated with a monoclonal anti-V5 antibody diluted 1:500 (Invitrogen) followed by anti-mouse IgG-A488 (Molecular Probes) diluted 1:1000. Cell nuclei were stained with DAPI (Calbiochem). Images were acquired in a Leica DMI6000B automated inverted microscope equipped with a Hamamatsu Orca R2 digital camera.

**Protein binding to cell surface by flow cytometry.** CHO-K1 cells were detached with 4 mM EDTA at 37 °C and harvested in PBS. Cells ($3\times10^5$ per experimental point) were incubated for 30 min on ice with 250 nM of the indicated viral recombinant proteins. Cells were then extensively washed with FACS buffer (PBS, 0.01% sodium azide and 0.5% bovine serum albumin) and incubated for 30 min at 4 °C with monoclonal anti-V5 antibody (Invitrogen) diluted 1:500 followed by anti-mouse IgG-A488 (Molecular Probes) diluted 1:500 in PBS. Finally, cells were resuspended in 0.5 ml FACS buffer and analysed in a FACSCalibur flow cytometer (BD Sciences).

**SPR analyses.** To determine the affinity constants, recombinant proteins (E194 and E194$^{GAGmut}$) were immobilized on CM4 chips (GE Healthcare) by amine coupling, as previously described[62], in order to get 350–800 response units. SPR experiments were performed using a Biacore X biosensor (GE Healthcare). Different concentrations (in the range 0.5–40 nM) of mouse IFNα subtype A (PBL Assay Science) were injected at 30 μl/min in HBS-EP and the collected sensorgrams were aligned and fitted to a 1:1 Langmuir model using the BIAevaluation 3.2 software.

**Neutralisation of type I IFN antiviral activity assay.** The two types of assay used in this work to measure IFN antiviral activity have been previously described[18,21]. In the preincubation assay, 50 units of recombinant murine IFN-α subtype A (PBL Assay Science) were incubated with supernatants from infected cells containing viral proteins, purified viral recombinant protein or diluted plasmas from infected mice for 30 min at 37 °C and then added to L929 cells seeded in 96-well plates the day before (2500 cells/well). In the washing assay, culture supernatants or recombinant proteins were added to cells, incubated for 30 min at 37 °C, and then extensively washed prior to the addition of murine IFN-α. In both cases, 16 h after IFN addition, cells were washed and infected with VSV (m.o.i. 50 pfu/cell) and cell viability was determined at 72 hpi using the Cell Titer 96 Aqueous One Solution cell proliferation assay (Promega) following manufacturer indications.

Supernatants from infected cells were collected at 72 dpi when cytopathic effect was complete. Supernatants were centrifuged at low speed to remove detached cells and cell debris and then inactivated by incubation with 2 μg/ml psoralen (4–9-aminomethyl-Trioxsalen; Sigma) for 10 min and then UV-irradiated for 10 min with 2.25 J/cm$^2$ in a Stratalinker 1800. Complete inactivation (>$10^8$ fold reduction in pfu) was confirmed by plaque assay on BSC-1 cells. Virus inactivated supernatants were concentrated 20–30 fold using Amicon ultra/filtration devices (Millipore) with a 10 kDa cut-off.

**Transcription analysis (RNA-seq).** Samples of the indicated tissues from two groups of five Balb/c mice per condition were aseptically removed at 5 dpi and conserved at 4 °C in RNAlater stabilisation solution (Qiagen). Total RNA extraction was performed with Reliaprep RNA tissue miniprep system (Promega) following manufacturer indications, and both quality and integrity of the RNA samples were assessed with the Agilent 2100 Bioanalyzer (Agilent Technologies). Duplicated cDNA libraries per condition (one per animal group) were constructed with TruSeq RNA Sample Prep Kit v2 Set A (Illumina) using equal amounts of total RNA from five animals per library, strictly following the manufacturer's instructions. In the case of the lymph nodes, tissues were pooled prior to RNA extraction and two cDNA libraries were also generated per condition. Libraries were sequenced using TruSeq SBS Kit v3—HS (Illumina) on an Illumina Hiseq 2000 at the Max-Planck-Institute for Molecular Genetics, Berlin. The fastq files containing the reads, after quality assessment with package FastQC, were mapped to the mouse genome (build GRCm38 from *Mus musculus* C57BL/6 J strain) using Tophat v2.0.4 with default parameters[63]. Only those reads matching the mouse genome were considered in the differential gene expression analysis carried out with Cuffdiff (Cufflinks v2.1.0 software[63]). Differentially expressed genes displaying statistically significant alterations (p-value < 0.05 and fold change >2, using two replicates for condition) were used to determine those pathways that were affected by the viral infection. Pathway gene enrichment and upstream regulators analyses were performed with the Ingenuity Pathway Analysis (IPA) software (Qiagen). Gene expression heatmaps were generated with the R package Gplots.

**Statistical analysis.** Data were analysed using GraphPathD Prism software. Footpad swelling and % initial weight data were analysed using multiple *t* tests with false discovery rate (FDR) Q = 1% and Mann–Whitney *U*-test was used with data related to virus titres in organs. Analyses were performed up to times post-infection at which survival rates in the corresponding groups were above 50%.

## Data availability
The data that support the findings of this study are available from the corresponding author upon reasonable request. The viral genomic sequences reported, together with the fastq files containing the reads from the RNA-seq experiments have been submitted to the European Nucleotide Archive under reference number PRJEB26437. A Reporting Summary for this Article is available as a Supplementary Information file.

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

## Acknowledgements

We thank Rocío Martín, Carolina Sánchez and M. Carmen Fernández for excellent technical assistance and the Genomics and Next Generation Sequencing Service at Centro de Biología Molecular Severo Ochoa for their support. We also thank Dr. Alí Alejo for critical reading of the manuscript. This work was funded by the Spanish Ministry of Economy and Competitiveness and European Union (European Regional Development's Funds, FEDER) (grant SAF2015-67485-R) and by European Sequencing and Genotyping Infrastructure (Seventh Framework Programme under grant agreement n° 262055). B.H. was funded by a JAE postdoctoral contract (Spanish Research Council).

## Author contributions

B.H. and A.A. conceived and designed the research; B.H., J.M.A.-L. and I.M. performed most of the experiments; C.F. and S.S. contributed to the RNAseq experiments; L.S. and N.S. contributed reagents and transgenic mice, respectively; B.H. and A.A. wrote the manuscript. All authors discussed the results and commented on the manuscript.

## Additional information

**Competing interests:** The authors declare no competing interests.

