## [Peer Review File · Nature Communications]

REVIEWERS' COMMENTS:

Reviewer #1 (Remarks to the Author):

Poxviruses encode a number of soluble secreted proteins that act as decoy receptors for host cytokines with prominent roles in the antiviral response. These fascinating molecules have been the object of multiple studies. Most of them have reported their existence and/or their impact during infection. However, very few studies have addressed in detail the properties of these unique proteins from the biochemistry to an animal infection model. This manuscript uses the VACV and ECTV interferon-binding protein (IFNBP) to elegantly demonstrate that their ability to bind GAGs on the cell surface is an essential property in the biology of these proteins. Wild-type and mutated recombinant IFNBP are used to demonstrate that the ability to bind IFN and the ability to bind GAGs lie in different parts of the molecule and therefore that the phenotypes can be probed by mutagenesis. The authors then generate recombinant viruses carrying these validated alleles and show how the GAG-binding activity of the IFNBPs contributes to virulence in two models of infection; allows for viral dissemination; and suppresses the IFN response in infected tissues. This is a detailed, well-conducted study that provides a valuable, and somewhat unexpected, insight into poxvirus immune evasion and cytokine biology. The paper is well-written and flows nicely. One could argue that the only piece missing is the analysis of the immune cells recruited to the site of infection when IFNBP is absent or unable to bind GAGs.

Specific points:

- The authors propose that the main advantage of binding GAGs is the retention of IFNBP at the site of infection to enhance its IFN blocking function. I would like the authors to consider that binding GAGs can also interfere with IFN-mediated leukocyte chemotaxis that occurs at the site of infection, a model that has already been proposed for other secreted poxvirus immunomodulatory proteins. This possibility should be addressed / discussed in the manuscript.
- Last sentence of discussion needs re-writing.
- Figure 8, right top panel, should read VACV rather than ECTV.

Reviewer #2 (Remarks to the Author):

Review on NCOMMS-18-20055A-Z Hernandez et al.

This manuscript reports from the further characterization of the immunomodulatory function of the type I interferon binding protein (IFN α /BBP) encoded by the orthopoxviruses vaccinia virus (VACV) and ectromelia virus (ECTV). In addition to the binding of type I interferon IFN α /BBP can interact with glycosaminoglycans (GAG) at the cell surface and this study investigates the relevance of the cell surface GAG binding activity of this secreted viral protein. Mutant VACV and ECTV viruses expressing a variant IFN α /BBP that lacks GAG binding activity were significantly attenuated upon in vivo infection and the cell surface retention of IFN α /BBP appears important to modulate the type I interferon inhibitory activity of this viral immune evasion protein. This is an exciting and technically well-done study firstly demonstrating the functional relevance of the GAG-binding property of a soluble viral decoy receptor. The following points should be addressed.

Specific comments:

1. Introduction: typo "monkeypox virus".
2. Results / Supp Fig. 4: "unhealthy intermediate appearance", the wording in the text describing the findings of gross pathology in livers and spleens should be improved.
3. Results / Fig. 3a Comparable replicative capacity: High virus titers, an exponential growth over

72h and an unusual increase in virus titers (over 7X log₁₀ for VACV) was obtained in the multiple-step growth analysis of wt and mutant viruses. This data needs a second look.

4. Results Fig. 8: Label upper left panel replace "ECTV" by "VACV".

RESPONSE TO REVIEWER COMMENTS

NCOMMS-18-20055A-Z Hernáez et al.

Reviewer #1 (Remarks to the Author):

Poxviruses encode a number of soluble secreted proteins that act as decoy receptors for host cytokines with prominent roles in the antiviral response. These fascinating molecules have been the object of multiple studies. Most of them have reported their existence and/or their impact during infection. However, very few studies have addressed in detail the properties of these unique proteins from the biochemistry to an animal infection model. This manuscript uses the VACV and ECTV interferon-binding protein (IFNBP) to elegantly demonstrate that their ability to bind GAGs on the cell surface is an essential property in the biology of these proteins. Wild-type and mutated recombinant IFNBP are used to demonstrate that the ability to bind IFN and the ability to bind GAGs lie in different parts of the molecule and therefore that the phenotypes can be probed by mutagenesis. The authors then generate recombinant viruses carrying these validated alleles and show how the GAG-binding activity of the IFNBPs contributes to virulence in two models of infection; allows for viral dissemination; and suppresses the IFN response in infected tissues. This is a detailed, well-conducted study that provides a valuable, and somewhat unexpected, insight into poxvirus immune evasion and cytokine biology. The paper is well-written and flows nicely. One could argue that the only piece missing is the analysis of the immune cells recruited to the site of infection when IFNBP is absent or unable to bind GAGs.

RESPONSE:

We thank the review for highlighting the interest and relevance of our findings. We agree that analysis of immune cells recruited to sites of infection will be of interest. Xu et al. (J. Exp. Med. 2008 205:981) showed that infection with an ECTV mutant in the IFNBP caused and increased recruitment and activation of CD8+, CD4+ and NK cells into draining lymph nodes, indicating that inhibition of IFN by the viral IFNBP reduces the adaptive and innate response. Since we show that expression of a mutant form of the IFNBP unable to bind to the cell surface causes an increased IFN response (as observed with IFNBP mutant viruses) one would expect a similar activation of CD8+, CD4+ and NK cells in mice infected with a GAG-mutant IFNBP. We have added a sentence in the Discussion to mention this result (p.17).

Specific points:

- The authors propose that the main advantage of binding GAGs is the retention of IFNBP at the site of infection to enhance its IFN blocking function. I would like the authors to consider that binding GAGs can also interfere with IFN-mediated leukocyte chemotaxis that occurs at the site of infection, a model that has already been proposed for other secreted poxvirus immunomodulatory proteins. This possibility should be addressed / discussed in the manuscript.

RESPONSE:

Although the interaction of the IFNBP with GAGs will mask a reduced number of GAG-binding sites at the cell surface, we agree that masking GAG binding sites may affect the interaction of chemokines with cell surface GAGs and indirectly leukocyte migration. We have added a sentence in the Discussion to mention this possibility (p. 17).

- Last sentence of discussion needs re-writing.

RESPONSE: we have done it.

• Figure 8, right top panel, should read VACV rather than ECTV.

RESPONSE: we have done it.

Reviewer #2 (Remarks to the Author):

Review on NCOMMS-18-20055A-Z Hernáez et al.

This manuscript reports from the further characterization of the immunomodulatory function of the type I interferon binding protein (IFN α / β BP) encoded by the orthopoxviruses vaccinia virus (VACV) and ectromelia virus (ECTV). In addition to the binding of type I interferon IFN α / β BP can interact with glycosaminoglycans (GAG) at the cell surface and this study investigates the relevance of the cell surface GAG binding activity of this secreted viral protein. Mutant VACV and ECTV viruses expressing a variant IFN α / β BP that lacks GAG binding activity were significantly attenuated upon in vivo infection and the cell surface retention of IFN α / β BP appears important to modulate the type I interferon inhibitory activity of this viral immune evasion protein. This is an exciting and technically well-done study firstly demonstrating the functional relevance of the GAG-binding property of a soluble viral decoy receptor. The following points should be addressed.

Specific comments:

1. Introduction: typo “monkeypox virus”

RESPONSE: we have done it.

2. Results / Supp Fig. 4: “unhealthy intermediate appearance”, the wording in the text describing the findings of gross pathology in livers and spleens should be improved.

RESPONSE: We have extended the description of the pathology in livers and spleen in the legend to Suppl. Fig. 4.

3. Results / Fig. 3a Comparable replicative capacity: High virus titers, an exponential growth over 72h and an unusual increase in virus titers (over 7X log₁₀ for VACV) was obtained in the multiple-step growth analysis of wt and mutant viruses. This data needs a second look.

RESPONSE: Maximal viral titres calculated for VACV were 2.2xlog₁₀, not 7xlog₁₀. However, as suggested by reviewer, we have re-examined these data to find a mistake related to the volume of virus inoculum used in the virus titre calculation, exclusively from VACV plaque assays. This is a small difference (now maximal VACV titres are 1.1xlog₁₀), but we have now incorporated this modification to the graphs with no change in the results or conclusions.

4. Results Fig. 8: Label upper left panel replace “ECTV” by “VACV”

RESPONSE: we have done it.